# Health Benefits Related to Tree Nut Consumption and Their Bioactive Compounds

**DOI:** 10.3390/ijms22115960

**Published:** 2021-05-31

**Authors:** Teresa Gervasi, Davide Barreca, Giuseppina Laganà, Giuseppina Mandalari

**Affiliations:** 1Department of Biomedical and Dental Sciences and Morphofunctional Imaging, University of Messina, 98168 Messina, Italy; teresa.gervasi@unime.it; 2Department of Chemical, Biological, Pharmaceutical and Environmental Science, University of Messina, 98168 Messina, Italy; giuseppina.lagana@unime.it (G.L.); giuseppina.mandalari@unime.it (G.M.)

**Keywords:** tree nuts, glucose modulation, oxidative stress, inflammation, body weight management, cardiovascular, functional foods, cognitive function, aging

## Abstract

Long-term studies with regular tree nut consumption have indicated positive outcomes for multiple health benefits. Here, we review the beneficial effects of tree nuts, highlighting the impact on glucose modulation, body weight management, cardiovascular risk, inflammation, oxidative stress, cognitive performance, and gut microbiota. Nuts are important sources of nutrients and phytochemicals, which, together with a healthy lipid profile, could help prevent certain chronic diseases, protect against oxidative stress and inflammation, and improve cognitive performance, thus reducing the impact of aging and neurodegeneration.

## 1. Introduction

Tree nuts are considered high in essential nutrients, and their consumption is known to have a beneficial influence on health outcomes [1,2,3]. Since ancient times, nuts have been part of the human diet and have been used for their medicinal properties [4]. To date, as a result of the health benefits related with habitual nut intake, the consumption of nuts is promoted in many dietary guidelines all over the world. Nut intake, which in the last 20 years has increased considerably, is common in all human diets, especially of the Mediterranean area [5,6].

Almond (*Prunus dulcis* Mill. D.A. Webb) is the most consumed tree nut throughout the world. North America is the largest producer of almonds, pistachios (*Pistachia vera* L.), and walnuts (*Juglans regia*), while the Middle East countries produce mostly cashews (*Anacardium occidentale*) and hazelnuts (*Corylus avellana*) [7]. 

It is believed that the nutrients contained in tree nuts are responsible for their potentially beneficial influence on human health. Numerous in vitro, in vivo, clinical, and epidemiologic studies have associated nut intake with a wide range of health benefits, including the modulation of glucose level [8,9,10,11,12] and serum lipids [13,14,15,16,17,18,19], positive influence on body weight [17,20,21,22] as well as the intestinal microbiota [23,24,25,26,27,28], antioxidant and anti-inflammatory activities [12,13,29,30,31,32,33,34,35,36,37,38,39,40], and, consequently, protective effect against certain chronic conditions, such as diabetes [8], obesity [20,21], and cardiovascular diseases [19,41].

The increased chronic inflammation, oxidative stress, and vascular impairment are considered contributing factors for the development of dementia, which is a progressive condition leading to a drastic decline in various cognitive domains, including planning, working memory, and processing speed, as well as codification and executive functions [42]. A number of strategies are currently being evaluated in order to be able to reduce risk factors that could lead to dementia up to 30 years before the onset of the symptoms.

Research suggests that diet is an important factor affecting cognitive performance in healthy people and could therefore have an impact on dementia onset in communities and countries [43]. A recent systematic review has analyzed the effects of nut consumption on cognitive performance [44].

In the present review, we will summarize the health benefits associated with nut consumption, highlighting the effect on glucose modulation, body weight management, cardiovascular risk, inflammation, oxidative stress, cognitive performance, and gut microbiota. Furthermore, the bioactive compounds that may be involved in cognitive effects are analyzed. 

## 2. Nuts and Health

This review is aimed to report the health outcomes identified with nut consumption, which are summarized in five main categories: glucose modulation, body weight management, cardiovascular disease prevention, inflammation and oxidative stress, functional food properties, and the cognitive performance of nuts.

### 2.1. Glucose Modulation

Several studies highlighted the role of nuts consumption in modulating glucose homeostasis in healthy [12], pre-diabetic [8,10], diabetic [9], and obese [11] subjects (Table 1). The beneficial role in the glucose and insulin metabolism of nuts has been associated with their content in fiber, fat, minerals, and other bioactive molecules. It has been shown that the fiber and fat content of almonds [10,12], the PUFA content of walnuts [11], and the PUFA content and the procyanidins, γ-tocopherol, and carotenoids such as lutein and β-carotene of pistachio [9,13] are responsible for these positive outcomes.

Jenkins et al. reported on the influence of almond intake on glycemic control. Fifteen 12 h fasting healthy subjects (aged 19–51, BMI >17.4 and <29.5 kg/m^2^) were assigned to five different dietary sessions, with a one-week wash-out period in between. Each session included four different test meals, each containing 50 g of available carbohydrates: two bread control meals and three test meals: almonds (60 g) and bread, parboiled rice, and instant mashed potatoes, which were balanced in carbohydrates, fat, and protein. The almond-containing meal promoted satiety (the incremental response area was higher than the control after 2 h and 4 h, *p* = 0.047 and 0.011, respectively), and it positively affected postprandial glycemia (glycemic indices of the almond (55 ± 7) and rice meals (38 ± 6) were lower than that of the instant mashed potato meal (94 ± 11) and the postprandial glucose peak heights for the almond (5.9 ± 0.2 mmol/L) and rice meals (5.8 ± 0.1 mmol/L) were less than the peak height for the potato meal (6.6 ± 0.2 mmol/L) and the control white bread meal (6.9 ± 0.2 mmol/L) (*p* < 0.001).

The insulin indices of the almond (91 ± 19) and rice meals (73 ± 9) were lower than the potato meal (171 ± 19) (*p*< 0.001), and the insulin peak height for the almond (224 ± 24 pmol/L) and rice (239 ± 28 pmol/L) meals were both lower than the potato meal (388 ± 30 pmol/L) (*p* < 0.001) and the control white bread meal (321 ± 36 pmol/L) (*p* < 0.042). Furthermore, the changes in the protein thiol concentration (15 ± 14 mmol/L following the almond meal and −10 ± 8 mmol/L after the control bread, rice, and potato meals) demonstrated that the almond-containing meal positively influenced oxidative stress [12].

A similar study was conducted on 137 individuals with type 2 diabetes mellitus (T2DM) risk (aged 18–60, BMI ≤ 27 and <18.5 kg/m^2^) over a 4-week period. The intake of 43 g/d of almonds promoted a lower serum glucose concentration after 60 min ingestion compared with the control (no almond supplementation). According to the authors, the moderating effects of almond consumption on postprandial glycemia may be attributable to the fiber and fat content of almonds. Fiber reduces glycemia by increasing the viscosity of intestinal contents and thus hindering glucose diffusion, by reducing carbohydrate availability in the gastrointestinal tract, and by starch encapsulation. Fat may have also caused lower postprandial glycemia by slowing gastric emptying times and dilution. In addition, serum glucose concentrations decreased when almonds were consumed alone as snacks, suggesting an enhancement of clearance. Furthermore, almond consumption positively affected satiety during the acute-feeding session, increased dietary of monounsaturated fat and α-tocopherol and, over a 4-week period, did not affect body weight or postprandial lipid profiles [10]. Gulati et al. also demonstrated that that inclusion of almonds in a healthy balanced diet had multiple beneficial effects on glycemic and CVDs risk factors amongst Asian Indian patients suffering from type 2 diabetes [45].

Pistachio intake has also been shown to confer positive metabolic effects on prediabetic [8] and diabetic [9] individuals. In a crossover study, 54 prediabetic individuals (aged 25–65, BMI < 35 kg/m^2^) consumed two diets, a pistachio-supplemented diet (PD) and a control diet (CD), each for 4 months. A 2-week washout period separated study periods. Diets were isocaloric and matched for protein, fiber, and saturated fatty acids. A total of 55% of the CD calories were derived from carbohydrates and 30% were derived from fat, whereas 50 and 35% of carbohydrates and fat, respectively, were included in the PD diet (with 57 g/day of pistachios). Pistachio intake promoted a significant decrease in fasting glucose (FBG) (mean −5.17 and 6.72 respectively for PD and CD; *p* < 0.001), insulin (mean −2.04 and 2.51 respectively for PD and CD; *p* < 0.001), and Homeostatic Model Assessment for Insulin Resistance (HOMA-IR) (mean −0.69 and 0.97 respectively for PD and CD; *p* < 0.001). Furthermore, pistachios showed beneficial effects on the inflammatory and oxidative state. According to the authors, the benefits for glucose metabolism and cardiovascular health promoted by pistachios are due to their higher amount of polyunsaturated fatty acids (PUFA) and other bioactive compounds, including procyanidins, γ-tocopherol, and carotenoids such as lutein and β-carotene [8]. 

In addition, diabetic patients have been shown to benefit from the consumption of pistachios. Forty-eight patients with type 2 diabetes were equally assigned to two groups (aged 53 ± 10 and 50 ± 11, BMI 32.16 ± 6.58 and 3024 ± 4.03 kg/m^2^, respectively). One group received a snack of 25 g pistachio nuts twice a day for 12 weeks and the second received a control meal without nuts. After 12 weeks intervention, patients had an 8-week washout and the groups were exchanged. A decrease in glycosylated hemoglobin A1 (HbA1c) (−0.4%) and FBG concentrations (−16 mg/dl) in the pistachio group compared with the control group (*p* ≤ 0.001 for both) were observed. Analysis of the two phases separately shown that pistachio consumption reduced systolic blood pressure (*p* = 0.007), BMI (*p* = 0.011), and CRP (*p* = 0.002) in patients from the treatment groups. However, no overall significant differences between the two groups in terms of BMI, blood pressure, HOMA-IR, and C-reactive protein (CRP) concentrations have been reported. According to the authors, these results are attributed to the washout period, which may have been too short [9]. Kendall et al. assessed the effect of pistachios on postprandial glucose and insulin levels, gut hormones, and endothelial function: results demonstrated that pistachios consumption reduced postprandial glycemia, increased glucagon-like-peptide levels, and could have insulin-sparing properties [46]. 

A study was carried out on 50 overweight adults with type 2 diabetes mellitus (mean age 54 ± 8.7 years, and BMI > 25 and <32 kg/m^2^) over a 1-year period who were randomized into two groups: control and diet. The control diet comprised of 30% fat (10% saturated fatty acids (SFA), 15% monounsaturated fatty acids (MUFA), 5% PUFA, and a P/S ratio of 0.5), 20% protein, and 50% carbohydrates, whereas the walnut diet included 30 g of walnuts per day (which provided 10% MUFA, 10% PUFA, and a P/S ratio of 1.0). Within the first three months, the walnut diet promoted an increased dietary PUFA and greater reduction in fasting insulin levels (*p* = 0.046) and body weight (which was significantly different from baseline, *p* = 0.028). The authors asserted that walnuts induced significant effects in the early stages of nutritional change in a delivered PUFA healthy diet, although long-term effects were subjected to fluctuations, which may be due to dietary intake and the disease process [11]. 

### 2.2. Body Weight Management

A large number of studies reported no adverse effects of nut consumption on energy balance or body weight [11,17,20,21,22] (Table 2). In addition, nut consumption, thanks to both the content in unsaturated fatty acids, dietary fiber, plant protein, antioxidants, vitamin E, arginine, phytosterols, and minerals such as potassium, calcium, and magnesium content and the non-bioaccessible nutrients, which presents prebiotic properties, has been shown to be inversely associated to metabolic syndrome (MetS) and excess weight [10,20].

A 24-week randomized controlled trial in 95 overweight individuals consuming mixed tree nuts resulted in both weight loss and increased satiety, together with decreased heart rate, and increased serum oleic acid after 24 weeks [47]. 

Through a meta-analysis, despite the heterogeneity of the observations, Li et al. highlighted the beneficial influence of nuts on the metabolic profile, analyzing the association of nut consumption with MetS and overweight/obesity. This meta-analysis of prospective cohort studies included six prospective cohort studies with 420,890 subjects and 62 randomized feeding trials with 7184 participants. The study showed that for every 1-serving/week increase in nut consumption, the risk was reduced by 4% for Mets, by 3% for overweight/ obesity, and by 5% for obesity only. Furthermore, pooled data of randomized feeding trials suggested that nut supplementation was related to a reduction in body weight (weighted mean difference (WMD): −0.22 Kg, 95% CI: −0.40 to −0.04), body mass index (WMD: −0.16 Kg/m^2^, 95% CI: −0.31 to −0.01), and waist circumference (WMD: −0.51 cm, 95% CI: −0.95 to −0.07). According to the authors, the observed benefits of nuts are due to the abundant bioactive compounds including unsaturated fatty acids, dietary fiber, plant protein, antioxidants, vitamin E, arginine, phytosterols, and minerals such as potassium, calcium, and magnesium. These healthy nutrients, alone or in combination, may improve inflammatory response, oxidative stress, and endothelial function, thus contributing to the improvement in individual MetS component. In addition, the non-bioaccessible nutrients from nuts (polymerized polyphenols, polysaccharides, and fiber), which have prebiotic properties, play an important role. The weight-loss effect of nuts was likely to be related to the enhanced satiety, the increased resting energy expenditure, the diet-induced thermogenesis, and the incomplete mastication and fat malabsorption [20]. 

Recently, a randomized crossover study reported that the energy content of almonds may less bioaccessible in individuals with hyperlipidemia than what was predicted by Atwater factors [48]. These findings support previous investigations on the role played by food processing and structure on the metabolizable energy of almonds [49,50]. 

Dhillon and co-authors showed the positive influence of almond intake on overweight and obese individuals [21]. Eighty-six overweight or obese adults (aged 18–60, BMI > 25, and <40 kg/m^2^) were randomly assigned, for a 12-week period, to 2500-kcal deficit diets: an almond-enriched diet (AED), in which 15% energy come from almonds, or a nut-free diet (NFD). The almond intake promoted a reduction in truncal fat (AED: −1.21% ± 0.26%; NFD: −0.48% ± 0.24%; *p* = 0.025), total fat (AED: −1.79% ± 0.36%; NFD: −0.74% ± 0.33%; *p* = 0.035), diastolic blood pression BP (AED: −2.71 ± 1.2 mm Hg; NFD: 0.815 ± 1.1 mm Hg; *p* = 0.029) and in visceral adipose tissue (VAT) loss (AED: −8.19 ± 1.8 cm^2^; NFD: −3.99 ± 1.7 cm^2^; *p* = 0.09). The authors suggested that moderate almond consumption may contribute to reduce metabolic disease risk in obesity [21].

In a long-term diet intervention trial, carried out over 18 months, conducted by Foster et al. on 123 overweight and obese subjects (46.8 ± 12.4 BMI: 34.0 ± 3.6 kg/m^2^), randomized into AED (28 g almonds daily) or NFD diet intervention groups, no statistically significant differences were shown between the two groups in terms of weight loss, body composition, or blood pressure, although the AED group experienced greater improvements in lipid profiles [17].

In addition, the effects of chronic peanut consumption on energy balance and body weight have been evaluated. Fifteen healthy, normal-weight participants (aged 33 ± 9, BMI = 23.3 ± 1.8 kg/m^2^) during a 30-week period were sequentially assigned to three different arms: the free-feeding (FF) arm was an 8-week trial where 50% of dietary fat energy was supplied by peanuts and the background diet was not controlled; the addition (ADD) arm was a 3-week trial including an isocaloric diet, in which 50% of dietary fat energy was supplied by peanuts; the substitution (SUB) arm was an 8-week trial where participants decreased fat intake by 50%, and this was replaced with an equivalent amount of fat from peanuts. Washout periods between each arm of the study were 4 weeks. During FF, total daily energy intake was significantly lower than predicted. The mean energy compensation score was 66%. Body weight gain (1.0 kg) was significantly lower than predicted (3.6 kg; *p* < 0.01). When customary dietary fat was replaced with the energy from peanuts, energy intake, as well as body weight, were maintained precisely [22].

In a randomized controlled study with non-diabetic overweight/obese adults, regular consumption of pistachios was associated with weight loss and reductions in BMI and waist circumference [51]. Furthermore, daily intake of pistachios (44 g) improved nutrient intake without affecting body weight in healthy women [52]. 

### 2.3. Cardiovascular Disease Prevention and Serum Lipid

Table 3 reports the cardiovascular disease prevention and serum lipid effects related to nut consumption.

Epidemiological studies have consistently proven an association between nut intake and reduced risk of CVD [19], ischemic heart disease (IHD), and CVD incidence and mortality [41]. Nut consumption may not only offer protection against heart disease but also increased longevity [55,56].

As reported, nuts, due to their dietary fiber content, magnesium, PUFA fats, vitamin E, folic acid, flavonoids, polyphenols, and L-arginine, may play an important role in reducing the cardiovascular risk through multiple mechanisms: by having a positive influence on the glucose [8,9,10,11,12] and/or lipid homeostasis [13,14,15,16,17,18,19], obesity [11,17,20,21,22], hypercholesterolemia [14,19], MetS or T2DM [9,10,11,19,20,38], blood pressure [9,21,33], oxidation biomarkers and antioxidant defenses [8,9,18,29,37,38,40], flow-mediated dilatation [53,54], lipid [15,16,40], or DNA modification [34] and inflammatory status [30,31,32,33,34,35,36,39].

In addition, nut consumption was shown to significantly improve endothelial function, which is an important risk factor for CVD. A recent systematic review and meta-analysis reported the improved flow-mediated dilatation (FMD) upon nut consumption and highlighted that amongst other nuts considered (almonds, pecans, peanuts, hazelnuts, pistachios, cashews, macadamia nuts, and soy nuts), walnuts significantly increased the FMD in comparison with control (weighted mean difference: 0.40%). According to the authors, the different nutrient profiles of walnuts, consisting of higher amounts of ω-3 fatty acid, α-linolenic, and α-tocopherol may, at least partially, explain why walnuts markedly improved FMD. In addition, the authors highlighted the important role of L-arginine in improving endothelial function in subjects with impaired nitric oxide (NO) production—for example, patients with hypercholesterolemia or congestive heart disease [53].

The positive effects of nuts consumption on various cardiovascular disease risk factors included improvements in triglycerides [13,17], total cholesterol (TC) [13,14,17], and lipoprotein cholesterol [13,14,15,17].

In vivo experimental mouse models, in which animals were assigned to three groups (n = 10) based on initial body weight, were fed either an isocaloric control diet (no nuts), 8.1% pistachio diet (single nut), or 7.5% mixed nut diet (almonds, brazil nuts, cashews, macadamia nuts, peanuts, pecans, pistachios, and walnuts) for 8 weeks. Pistachios and mixed nuts significantly decreased triglycerides, TC, and LDL-C (*p* < 0.05) compared with controls and exhibited reductions in C-reactive protein (*p* = 0.045) and oxidative stress (*p* = 0.004). Furthermore, in the mixed nut group, the activities of the antioxidant enzymes, such as superoxide dismutase (*p* = 0.004) and catalase (*p* = 0.044), were significantly higher in the mixed nut group compared to the control group, and the aspartate aminotransferase (*p* = 0.048) concentration was lower than the control group. The authors concluded that mixed nuts and individual nut varieties have comparable effects on CVD risk factors in rats [13].

Forty-eight individuals (aged 30–65, BMI ≥ 20 and ≤35 kg/m^2^) with elevated LDL-C (149 ± 3 mg/dL) were enrolled and randomized in two 6-week feeding settings: a cholesterol-lowering diet (51% carbohydrates, 16% proteins, 32% total fat) supplemented with almonds (about 42.5 g of almonds/day) and a control diet (58% carbohydrates, 15% proteins, 26% total fat) supplemented with an isocaloric muffin substitution (no almonds/day). The differences in the nutrient profiles of the two diets were due to nutrients inherent to each snack; diets did not differ in saturated fat or cholesterol. The almond diet significantly decreased non-HDL-C (−18 ± 3 versus 11 ± 3 mg/dL; *p* = 0.01); *-* LDL-C (−19 ± 2 versus −14 ± 2 mg/dL; *p* = 0.01), TC/HDL-C (−0.17 ± 0.08 versus 0.06 ± 0.08; *p* < 0.01), LDL-C/ HDL-C (−0.23 ± 0.07 versus −0.03 ± 0.07; *p* < 0.01), and apoB/apoA1 (−0.04 ± 0.01 versus −0.00 ± 0.01; *p* < 0.01) ratios. The abdominal mass (−0.28 ± 0.09 versus −0.09 ± 0.09 kg; *p* = 0.02) and abdominal fat mass (−0.13 ± 0.03 versus −0.06 ± 0.03 kg; *p* = 0.02) as well as the leg fat mass (−0.26 ± 0.06 versus −0.14 ± 0.06 kg; *p* = 0.02) were significantly reduced with the almond diet. These findings were validated by waist circumference, which also decreased with the almond diet (−1.7 ± 0.4 versus −0.9 ± 0.4 cm; *p* = 0.02) compared with the control diet. 

The authors attributed the cardioprotective effect of almonds, in part, to their unique fatty acid profile, which is high in unsaturated fat, predominantly oleic acid, and low in saturated fat, and suggested that other nutrients and bioactive compounds in almonds, such as dietary fiber and phytosterols, may contribute to their LDL-C-lowering and HDL-C-conserving effects [15]. Berryman et al. (2017) have later reported that the incorporation of almonds in a cholesterol-lowering diet improved plasma HDL and cholesterol efflux to serum in normal-weight participants with high LDL cholesterol [57].

These results are in line with a previous study, where significantly greater reductions in TC and an improvement in the ratio of TC to HDL cholesterol were observed after a 6-month intervention [17]. 

A recent randomized controlled trial demonstrated that almond snacking may reduce the CVD risk by increasing heart rate variability during mental stress in healthy adults, and whole almond consumption was associated with better diet quality and lower cardiovascular disease risk factors in the UK adult population [58,59]. The same authors have also reported a significant improvement on endothelial function associated to consuming whole almonds as snacks, in addition to LDL cholesterol lowering amongst adults with above-average risk of CVD [54]. In addition, pistachio consumption has been investigated on plasma lipid profiles in the parallel-design study by Kocyigit et al. Forty-four healthy individuals, equally randomized into two groups (mean age 32.8 ± 6.7, BMI 24.6 ± 5.6 and 33.4 ± 7.2, BMI 24.2 ± 6.1 kg/m^2^, respectively) were assigned to 3-week diet interventions: regular diet and whole-pistachio diet (20% of daily caloric intake). After the intervention, the mean plasma TC, malondialdehyde (MDA) levels, and TC/HDL and LDL/HDL ratios were found to be significantly decreased (*p* < 0.05, *p* < 0.05, *p* < 0.001 and *p* < 0.01, respectively) in the pistachio group. On the other hand, HDL levels, antioxidant potential (AOP), and AOP/MDA ratios were significantly increased (*p* < 0.001, *p* < 0.05, and *p* < 0.01, respectively). These results indicated that pistachio consumption not only improved TC and HDL levels but also decreased oxidative stress [18].

A recent study involving 54 subjects (aged 25–65, BMI 35 kg/m^2^) with prediabetes who followed two crossover sequences, a pistachio-supplemented diet (PD, 50% carbohydrates, 33% fat, including 57 g/d of pistachios daily) and a control diet (CD, 55% carbohydrates, 30% fat) for 4 months each, separated by a 2-week washout, confirmed the positive effects associated with nut consumption and showed the ability of nut intake to shift the lipoprotein size and particle profile to a less atherogenic pattern. Diets were isocaloric and matched for protein, fiber, and saturated fatty acids. Chronic intake of 57 g of pistachio decreased small low-density lipoprotein particles (sLDL-P), non high-density lipoprotein particles (non HDL-P), and the mean size of high-density lipoprotein particles (HDL-P), despite the absence of change in TC, LDL-C, or HDL concentrations. According to the authors, independently of changes in the total plasma lipid profile, pistachios may play a beneficial role in cardiovascular disease. It is known that small, dense LDL particles are responsible for greater atherogenic risk than large LDL particles. This is due to their interaction with the arterial wall. Furthermore, high levels of small, dense LDL have been positively correlated with microalbuminuria, with more non-calcified plaque and an atheroprotective role, whereas they were negatively correlated with glomerular filtration rate as predictors of diabetic nephropathy [16].

In addition, walnut supplementation increased the HDL after a 6-month treatment, which was possibly due to the higher PUFA intakes, although no long-term effects have been shown [11]. Cashew supplementation (from 28 to 64 g/day) in mildly hypercholesterolemic adults (aged 21–79, BMI ≥ 18, and <32 kg/m^2^) following a typical American diet (50% carbohydrates, 18% protein, 32% fats) decreased TC (−3.9% versus 0.8%, *p* = 0.005), LDL-C (−4.8% versus 1.2%, *p* = 0.008), non-HDL cholesterol (−5.3% versus 1.7%, *p* = 0.007), and total cholesterol/HDL cholesterol ratio (−0.0% versus 3.4%, *p* = 0.035), in comparison with a control diet, consisting of potato supplementation (54% carbohydrates, 18% protein, and 29% fat) [14].

Results of a systematic review by Mukuddem-Petersen et al. showed that the consumption of moderate-fat diets (35% of energy) containing ca. 50–100 g/d of nuts, especially almonds, peanuts, pecan nuts, or walnuts, significantly lowered TC and LDL-C concentrations compared with subjects consuming control diets. In particular, a diet supplemented with almonds (50–100 g/d), peanuts (35–68 g/d), pecan nuts (72 g/d), and walnuts (40–84 g/d) resulted in a decrease in total cholesterol between 2% and 16% and LDL cholesterol between 2% and 19% compared to an unsupplemented control diet. The consumption of macadamia nuts (50–100 g/d) produced less significant results. The authors concluded that the consumption of about 50–100 g of nuts approximately 5 times per week as part of a healthy diet with a total fat content (high in MUFA and/or PUFA) of about 35% energy may significantly decrease total cholesterol and LDL cholesterol in normo- and hyperlipidemia individuals [19].

### 2.4. Effect on Inflammation and Oxidative Stress

Nuts are sources of tocopherols and phenolic compounds with potent antioxidant and anti-inflammatory properties. A large number of in vitro [34], in vivo [30,31,32,33,34,35,36,39], and epidemiological [8,12,29,37,38,40] studies have highlighted the beneficial effects of nut intake on inflammatory and oxidative processes (Table 4).

Recently, the potential beneficial effects of cashew nuts on chronic and acute inflammatory and oxidative processes have been investigated in different in vivo experimental mouse models, such as dinitrobenzene sulfonic acid (DNBS)-induced colitis and monosodium iodoacetate (MIA)-induced osteoarthritis, carrageenan-induced paw edema, cerulean-induced pancreatic and lung injury, and ischemia/reperfusion (I/R) injury [30,31,32,33,35]. It has been shown that cashew nuts intake at a dose of 100 mg/kg (1) alleviated pain perception and histological damage, (2) reduced various pro-inflammatory pathways and molecular mediators, including myeloperoxidase (MPO) and MDA levels, mast cell degranulation, neutrophil infiltration, and the release of pro-inflammatory cytokines, (3) modulated NF-κB signaling and ROS generation, (4) enhanced Nrf2 pathway activation, and (5) suppressed the NLRP3 pathway [30,31,32,33,35]. Cordaro et al. (2020) highlighted the anti-inflammatory, anti-oxidative, and analgesic properties of cashews and hypothesized a correlation between these beneficial effects and the high content of phenols, which mediate the activation of 5-LOX COX pathways [30]. Fusco et al. (2020b) also demonstrated how cashew nut intake decreased the intestinal barrier dysfunction and mucosal damage and the translocation of toxins and bacteria, which are usually correlated to systemic inflammation and organ injuries, particularly of liver and kidney [33].

In vivo studies have also demonstrated the effectiveness of pistachio, Brazil nut, and mixed nuts against oxidative stress [34,36] and inflammation [36]. 

Paterniti et al., using the CAR-induced histological paw damage mouse model, showed that treatment with polyphenols-rich extract obtained from natural pistachio (NP) significantly reduced the paw edema, with a decrease in MPO activity, as opposed to roasted pistachio (RP), which did not demonstrate a significant effect. According to the authors, the bioactivity may be due to the synergistic interaction amongst the polyphenols identified in NP, to the effect of catechin, which is present in higher concentration in NP compared to RP, as well as the effect of epicatechin and isoquercetin, which are also significantly higher in NP compared to RP [36]. Moreover, Lorenzon dos Santos et al. highlighted the effect of the consumption of different nuts on oxidative stress biomarkers, which are typically involved in the primary and secondary prevention of cardiovascular disease. The potential of nuts in exerting antioxidant effects by DNA repair mechanisms, lipid peroxidation prevention, modulation of the signaling pathways, inhibition of the MAPK pathways, suppression of NF-kB, and activation of the Nrf2 pathways was proven using in vitro tests. The ability of dietary nuts in improving biomarkers of oxidative stress, such as oxLDL and GPx was shown using animal models [34]. 

Overall, these results suggest that nut intake may contribute to protection against oxidative stress and the related consequences of inflammatory processes, which are mostly due to the abundance of secondary metabolites such as phenol compounds, flavonoids, and carotenoids, which are present in the skin and in the kernel, as shown in previous studies [39,60].

The antioxidant potential of almond consumption has also been investigated through a randomized crossover study involving 14 (12 h fasting) healthy subjects (aged 19–65, BMI ≥ 25 kg/m^2^) randomized into three 3-week crossover designs with a 1-week washout period between treatments: walnut (75% of energy intake), almond (75% of energy intake), and control (macronutrient composition similar to the nut containing meal). In both treatment groups, plasma polyphenol concentration significantly increased within 30 min after ingestion of walnuts and almonds, reaching peak levels at 90 min (224.3 ± 2.57 mg L^−1^ gallic acid equivalent (GAE) for walnut diet and 238.48 ± 2.71 mg L^−1^ GAE for almond meal). The antioxidant capacity reached a peak value at 150 min and started to decline at 210 min. At 150 min after the consumption of either the walnut and almond meal, both lipophilic and hydrophilic oxygen radical absorbance capacity (ORAC) components showed an increase in plasma antioxidant capacity over baseline (*p* < 0.05) (455.3 ± 3.8 and 130.5 ± 3.4 μmol L^−1^ ORAC lipophilic and hydrophilic for walnut diet and 472.5 ± 3.8 and 266.2 ± 8.7 μmol L^−1^ ORAC lipophilic and hydrophilic for almond meal). A gradual significant (*p* < 0.05) reduction in the susceptibility of plasma to lipid peroxidation was observed 90 min after ingestion of the nut meals (6.4 ± 0.9 μmol L^−1^ MDA for walnut diet and 5.2 ± 0.9 μmol L^−1^ MDA for almond meal). The results for the control meal showed no significance [40].

Epidemiological and clinical studies suggested that some dietary factors, such as ω-3 PUFA, antioxidant vitamins, dietary fiber, L-arginine, and magnesium may play an important role in modulating inflammation. The relationship observed between frequent nut consumption and inflammatory markers has been investigated in controlled feeding intervention and has reported mixed results [8,29,37,38]. 

Recently, the potential of nuts, when used in controlled amounts in a weight management program, to promote weight reduction and improve the plasma concentration of inflammatory factors was shown in a randomized controlled parallel trial. Sixty-seven overweight and obese stable coronary artery disease individuals (age 58.8 ± 7.4 years, BMI 30.9 ± 3.9 kg/m^2^) were randomly allocated to an 8-week nut-enriched low-calorie diet (NELCD) or to a nut-free low-calorie diet (NFLCD). The NELCD promoted a decrease in ICAM-1 (*p* = 0.04) and, IL-6 (*p* = 0.02) concentrations compared to NFLCD. No significant difference in concentrations of monocyte chemo-attractant protein (MCP-1) or plasma CRP was observed between diet groups [29].

Liu et al. showed that almond consumption improved inflammation and oxidative stress in 20 Chinese patients (age 58 ± 2 years, BMI 26 kg/m^2^) with T2DM and mild hyperlipidemia. Within a 12-week randomized, crossover, controlled feeding trial, after a 2-week run-in period, individuals were assigned to receive either a control (56% carbohydrates, 17% protein, and 27% fat) or an almond diet (56 g/day were added to the control diet to replace 20% of total daily calorie intake) for 4 weeks each, with a 2-week washout between alternative diets. The almond diet decreased IL-6 (by a median 10.3% compared to the control diet—95% confidence intervals 5.2, 12.6%), CRP (by a median 10.3% compared to the control diet—24.1, 40.5), TNF-α (by a median 15.7% compared to the control diet—0.3, 29.9) and also the plasma protein carbonyl (PC) (by a median 28.2% compared to the control diet—4.7, 38.2), though it did not alter antioxidant capacity and total phenolic content in plasma and plasma MDA. The almond diet also enhanced LDL resistance against Cu2+-induced oxidation (by a median 16.3%—7.4, 44.3) as compared to the control diet. Serum intercellular adhesion molecule-1 and vascular adhesion molecule-1 were not changed by either diet [38].

In another study, Tey et al. have reported that the inclusion of hazelnuts into the usual diet did not significantly influence biomarkers of inflammation and endothelial function and blood lipid profiles. In this study, 107 weight-stable, overweight and obese participants (aged 18–65, BMI ≥ 25 kg/m^2^) followed a randomized, controlled, parallel 12-week intervention including three treatment arms: no nuts (control group), 30 g/d of hazelnuts, or 60 g/d of hazelnuts. With the exception of a tendency toward improvement in VCAM-1 concentration in the 60-g/d nut group (*p* = 0.07), no significant differences in follow-up clinical outcomes between groups were observed for the measured parameters, which included blood pressure, body composition, plasma high-sensitivity C-reactive protein (hs-CRP), IL-6, ICAM-1 [37]. The authors suggested that a dietary intervention with either energy restriction or weight loss may be necessary in order to improve obesity-related inflammatory markers, as shown by several studies, which reported improvements in these biomarkers [37].

In addition, pistachio intervention has been shown to determine a decrease of IL-6 mRNA and resistin gene expression by 9 and 6%, respectively (*p* < 0.05, for PD vs. CD) in lymphocytes. Furthermore, other cardiometabolic risk markers such as fibrinogen, oxidized LDL, and platelet factor significantly decreased under the PD compared with the CD (*p* < 0.05), whereas glucagon-like peptide-1 increased. According to the authors, the beneficial effects on the inflammatory and oxidative state may be due to the high content in lutein, β-carotene, and tocopherol in pistachios [8].

### 2.5. Functional Food Properties

Tree nuts, mainly almonds and pistachios, have a positive effect on the composition of the bacterial and fungal fecal microbiota (Table 5). 

Several studies pointed out the potential of nut or/and the nuts skin, especially almond skin, to be used as source of prebiotics [24,25,27], due to non-bioaccessible nutrients and phytochemicals in nuts such as polymerized polyphenols, ellagitannins, and proanthocyanidins, which are fiber that enter the colon intact [61].

Mandalari et al. showed that finely ground almonds (FG) subjected to a combined model of the gastrointestinal tract that included in vitro gastric and duodenal digestion, when used as substrates for the colonic model to assess their influence on the composition and metabolic activity of gut bacteria populations, increased the populations of bifidobacteria and *Eubacterium rectale*, resulting in a high prebiotic index. No effect was observed when using defatted finely ground almonds (DG) [25]. The authors highlighted the lipid component of almond seeds as a relevant element in the alteration of bacterial growth and metabolism, given that only a small amount of almond lipids and proteins are bioavailable during gastric and small intestinal digestion due to nutrient encapsulation by cell walls likely to prevent digestion in the upper GIT [26].

In another study using an in vitro gastric and duodenal digestion model, followed by colonic fermentation, the potential prebiotic effect of almond skins, by-products of industrial blanching, was also proven. The study showed the ability of natural (NS) and blanched (BS) almond skins to significantly increase in vitro the bifidobacteria and *Clostridium coccoides/Eubacterium rectale* numbers. This highlighted the beneficial effects of the nonglycemic carbohydrates, mainly pectin, present in almond skins and showed that polyphenols, present in almond skins, did not affect bacterial fermentation [27]. Furthermore, almond skins have been shown to have high fiber content as well as significant amounts of lipid; both of these components may be relevant to fermentation in the large intestine [28]. 

A very recent randomized study enrolling 102 participants with HIV demonstrated that a Mediterranean diet, supplemented with extra virgin olive oil and walnuts, may improve metabolic indicators, immune activation, and gut microbiota diversity for people affected with HIV-1 [62]. The authors showed an improved lipid profile, whereas the immune activation and IFN-γ-producing T-cells were reduced in the group supplemented with olive oil and walnuts.

Recently, the effect of pistachio consumption on the gut microbiota composition has been reported. The characterization of microbiota in fecal samples collected from volunteers who were recruited to participate in two separate randomized, controlled, crossover studies was carried out. The subjects were assigned to three treatment groups: no nuts; 1·5 servings/d either almonds or pistachios, and 3 servings/d of either almonds or pistachios; the feeding periods were separated by a washout period of at least 2 weeks. It has been shown that the effect of pistachio consumption on gut microbiota composition was stronger than almond and included an increase in the number of potentially beneficial butyrate-producing bacteria [24]. 

Other studies are consistent with these findings, showing how the abundance of dietary fiber and polyphenols may be correlated to the prebiotic effects of almond skins and almonds [23].

Liu et al. compared the daily supplementation of roasted almonds (56 g), almond skins (10 g) in the diet of 48 healthy adult volunteers, for 6 weeks, to a commercial fructooligosaccharides (8 g) supplementation. The almond and skin intake promoted significant increases in the populations of Bifidobacterium spp. and Lactobacillus spp. in fecal samples, while the growth of the pathogen *Clostridum perfringens* was significantly repressed and *Escherichia coli* numbers did not change significantly, compared to the control [23]. Dhillon et al. (2019) also demonstrated that almond snacking in 73 college freshmen for 8 weeks increased the alpha-diversity of the GI microbiome and significantly decreased *Bacteroides fragilis* abundance compared [63]. 

### 2.6. Cognitive Performance of Nuts 

Almonds are rich in nutrients that benefit cognitive function. A number of studies have identified an effect of almond consumption on cognitive performance (Table 6). 

A recent study [64] reported on the effects of daily almond consumption for six months on cognitive measures in healthy middle-aged to older adults through a randomized control trial: results confirmed that although serum markers of inflammation and oxidative stress were not significantly affected across the groups, there was a significant improvement in visuospatial working memory, visual memory and learning, and spatial planning and working memory in subjects receiving almonds. Furthermore, the serum alpha-tocopherol concentrations increased by 8% in the almond group only. Coates et al. examined supplementing habitual diets with either almonds or carbohydrate-rich snack foods on biomarkers of cardiovascular and metabolic health, mood, and cognitive performance in overweight/obese people, 50–80 years old [65]. While the almond-rich diet significantly reduced triglycerides, no other changes in cardiometabolic biomarkers, mood, or cognitive performance were detected. In a randomized trial with overweight and obese adults, Dhillon et al. observed that the consumption of an almond-enriched high-fat lunch ameliorated the post-lunch dip in memory, although it did not further enhance the cognitive function outcomes with weight loss over 12 weeks [66]. 

When 522 participants at high vascular risk from a primary prevention trial (PREDIMED) followed a nutritional intervention comparing two MedDiets (supplemented with either extra-virgin olive oil (EVOO) or mixed nuts) versus a low-fat control diet, higher mean Mini-Mental State Examination (MMSE) and Clock Drawing Test (CDT) scores were recorded with either EVOO or nuts. Therefore, nuts appeared to improve cognition compared with a low-fat diet [67]. In another study, the same authors reported no significant differences in cognitive performance and cognitive status (normal, mild cognitive impairment, or dementia) in participants assigned to the MedDiet+Nuts group [68]. Another randomized controlled trial to define the effect of Mediterranean diet on cognitive function enrolling older women identified no association between total nut intake and global cognitive function [69]. 

A cross-sectional study performed in Northern Italy involving 279 participants aged ≥ 65 years (80 men, 199 women) was carried out to determine adherence to the Mediterranean diet and its association with cognitive function [70]. Results showed that nut consumption was associated with a lower risk of cognitive impairment (OR = 0.30; 95% CI, 0.13–0.69, *p* = 0.005). Another Italian-based study enrolling 119 older participants concluded that nut consumption estimated either by the dietary marker or by the urinary marker model was associated with less cognitive decline (OR: 0.78, 95% CI: 0.61, 0.99; *p* = 0.043 and OR: 0.995, 95% CI: 0.991, 0.999; *p* = 0.016, respectively) with AUCs 73.2 (95% CI: 62.9, 83.6) and 73.1 (62.5, 83.7), respectively [71]. 

The Hordaland Health Study assessed cognitive performance among the elderly in relation to the intake of plant foods, including fruits, vegetables, potatoes, grain products, mushrooms, and nuts [72]. No significant differences were observed between groups after consumption of nuts, whereas subjects with intakes of >10th percentile of fruits, vegetables, grain products, and mushrooms performed significantly better in cognitive tests.

In an animal study employing healthy rats, repeated administration of almonds was shown to increase brain acetylcholine levels and enhance memory function. The memory deficits were also attenuated [73]. Another animal study evaluated the almond consumption in scopolamine-induced amnesia in rats [74]. The authors detected a significantly reversed scopolamine (1 mg/kg i.p.)-induced amnesia after 7 and 14 days of almond administration. The brain cholinesterase activity was also reduced after almond consumption, together with a cholesterol and triglyceride lowering property and a slight increase in glucose levels.

Taken together, these reports show a positive effect of almond consumption in animal studies and in certain clinical trials, indicating the need to define the impact of the study time frame, age of participants, and any pre-existing metabolic disorder. These factors, together with changes through dietary manipulation, may affect cognition measures.

A cross-sectional study on the relationship between walnut consumption and cognitive function indicated positive associations among all adults, regardless of age, gender, or ethnicity, suggesting that daily walnut intake could be a beneficial dietary behavior [75]. In particular, adults aged 20–59 years old required 16.4 ms less time to respond on the simple reaction time and 0.39 s less for the symbol digit substitution after consuming an average of 13.1 g walnuts per day. The effects of walnut consumption on cognitive performance in young adults has also been established by Pribis et al. through a randomized controlled trial: although no significant increases were detected for mood, non-verbal reasoning or memory as a result of the walnut-supplemented diet, inferential verbal reasoning increased significantly by 11.2% [76]. A recent study with a representative sample of 3632 US adults aged 65 years and older established an association between walnut consumption and cognitive function, with greater cognitive scores in volunteers consuming walnuts [77]. However, walnut consumption was not protective against age-related cognitive decline. Chauhan and Chauhan [78] have also recently reported on the beneficial effects of walnuts on cognition and brain health, highlighting the benefits of a walnut-enriched diet in brain disorders and in other chronic diseases. An animal study suggested that dietary supplementation with walnuts may have a beneficial effect in reducing the risk, delaying the onset, slowing the progression of, or preventing Alzheimer’s disease [79]. In another study with aged rats (19 months old), a diet supplemented with 6% walnuts reported to improve cognitive and motor performance [80].

Fewer studies are available in the literature on the cognitive performance related to pistachio consumption. However, significantly positive outcomes were detected in a number of animal studies [81,82]. In a recent study, Nuzzo et al. demonstrated that a regular intake of pistachios mitigated the effects of a high-fat diet (HFD) in the brain of obese mice: the pistachio diet significantly reduced the serum levels of triglycerides and cholesterol in the HFD model, whereas no significant differences were observed in insulin resistance between the two diets. Furthermore, the impaired mitochondrial function found in HFD brain was partially recovered after the pistachio diet, together with a decrease in reactive oxygen species, singlet oxygen, and phosphorylated extracellular signal-regulated kinase. A positive outcome was also revealed in male rats subjected to treatment with cisplatin or vincristine whose diet was supplemented with pistachio (10%) to attenuate motor and cognition impairments [82]. 

The potential of *Pistacia vera* fruits in experimental memory impairments was assessed in Swiss Albino mice whose memory impairment was induced by scopolamine [83]. Pretreatment of mice with an ethanolic fraction of *Pistacia vera* fruit significantly reduced scopolamine-induced amnesia, with an increase in escape latency and in step-down latency. The demonstrated cognitive improvement was related to treatment with *Pistacia vera* fruits. 

In a randomized controlled trial, the consumption of brazil nuts provided some evidence of positive cognitive effects on verbal fluency and constructional praxis compared with the control groups [84].

Given the limited literature data on cognitive performance of certain tree nuts, we believe it would be worthwhile to explore their potential in the management of memory disorders in humans.

#### Bioactives Related to Cognitive Function

The perfect functionality of our brain is perhaps one of the most important elements for the regulation of our body and its performance. The brain requires a high amount of energy, which is mainly supplied by glucose. Since neurons are particularly susceptible to oxidative stress, foods rich in omega-3 fatty acids, such as nuts, may help build and repair the brain damaged cells. Phenols, polyphenols, and vitamins also reduce cellular oxidative stress onset, inflammation, and activation of the apoptotic pathway, which could be directly linked to brain aging and neurodegeneration [85,86,87,88,89,90]. In general, seeds are a rich source of several “brain-friendly” compounds, including natural oils [86,91]. However, what are these active compounds, and how do they act to keep our brain sharp and in good health? Nuts are particularly rich in both MUFA and PUFA, phenols and polyphenols compounds, minerals and vitamins, such as vitamin E [92,93,94]. The junction between a balanced diet, an active life style, and the assumption of these compounds will have a role in boosting the brain performance and memory [69,95]. Although so far, literature studies are very limited, some interesting results can be found in epidemiological and experimental works (with both in vitro and in vivo data). Given the high amount of omega-3 fatty acids, vitamin E, and antioxidants, nuts can protect neurons from free radicals and other reactive oxygen and nitrogen species (ROS and RNS) and slow down cognitive decline. Moreover, other compounds, such as riboflavin and L-carnitine, may have an effect on brain function, since they are involved in the development of neural pathways [86,96]. The antioxidant and anti-inflammatory components in walnuts, which include flavonoids, phenolic acid (ellagic acid), melatonin, folate, vitamin E, selenium, proanthocyanidins, and ω-3 -linolenic acid have also been reported to act additively or synergistically to reduce the risk of age-related diseases [78]. 

Another class of compounds with a potential effect on cognitive function is carotenoids, such as lutein and zeaxanthin, whose chemical structure allows it to act as efficient singlet oxygen and peroxyl radical scavengers. Amongst tree nuts, lutein and zeaxanthin are found in pistachios. A randomized, double-masked, placebo-controlled trial involving supplementation with lutein/zeaxathin to evaluate the cognitive function of community dwelling older adults showed that participants receiving the supplementation had statistically significant increases in macular pigment optical density, improvements in complex attention and cognitive flexibility domains, as well as executive function domain [97]. Moreover, the supplementation improved composite memory in male participants. Lutein is a natural dietary antioxidant that could help maintain the brain structure by lowering the level of oxidative stress and related damages [98] as well as chronic inflammation. Indeed, it has been demonstrated that lutein and zeaxathin are well-known strong anti-inflammatory molecules [99]. Moreover, Johnson and colleagues showed that lutein can be identified in higher concentration in the brain compared with peripheral blood circulation, as assessed using matched brain tissue and serum samples in a group of centenarians [100]. The effects of these compounds have also been analyzed in several studies on young people. For instance, preliminary data showed that supplementation with lutein and zeaxathin increased systemic levels of brain-derived neural growth factor in younger individuals [101], with increased visual processing speed and reaction times [102,103]. The mechanisms underlining these effects are not yet clear; it has been only speculated that the effect can be due to an influence on processing speed and brain connectivity [104], perhaps by enhancing gap junctions between neurons.

Almond consumption may have an indirect effect on brain cognitive functions based on its effectiveness in weight loss [105,106]. Studies based on weight loss intervention including almonds also reported an improvement in memory and attention, due to weight loss, and reduction of the post-lunch dip in memory [65,66,107]. Kulkarni et al. found a direct effect of almond intake and scopolamine-induced amnesia in rats [74]. The oral administration of three doses (150, 300, and 600 mg/kg) for 7 and 14 consecutive days to the respective groups of rats resulted in the reversion of scopolamine (1 mg/kg i.p.)-induced amnesia, as evidenced by a decrease in the transfer latency in the EPM task and step-down latency in the passive avoidance task. Moreover, oral administration of almonds resulted in a reduction of brain cholinesterase activity in rats, together with a remarkable decrease of blood biochemical parameters, such as cholesterol and triglyceride lowering, and a slight increase in glucose levels. 

### 2.7. Miscellaneous 

The effects of almond consumption on skin lipids and wrinkles have been investigated in a randomized controlled pilot study enrolling 50 healthy postmenopausal females: the natural antiaging benefits of almonds were reflected on the significant reduction of wrinkles severity [108]. Another recent randomized controlled trial concluded that the daily consumption of almonds could contribute to the improvement of facial wrinkles and reduction of skin pigmentation in postmenopausal women with Fitzpatrick skin types I and II [109]. Furthermore, recent findings suggested that daily oral almond consumption was related to enhanced protection from UV photodamage in healthy Asian women [110]. These data support our previous investigation on the photoprotective effect of blanch water, which is a by-product of the almond processing industry [111].

Almonds are a rich source of magnesium, whose consumption from dietary sources has been shown to be beneficial in reducing cancer mortality [112]. Furthermore, a recent systematic review comprehensive of 49 clinical trials reported a beneficial effect of oral prescribed magnesium therapy against hypertension [113]. 

A number of studies have indicated that walnuts provide key nutrients, such as alpha-linolenic acid, which can help with the male reproductive system. A randomized controlled trial demonstrated that walnuts added to a Western-style diet improved sperm vitality, motility, and morphology [114]. These findings have been corroborated by some animal studies, supporting the evidence of effectiveness of walnuts on sperm quality [115,116].

A randomized controlled study has reported the beneficial effects of a Mediterranean diet with additional extra virgin olive oil and pistachios on gestational diabetes mellitus, whose prevalence is increasing and becoming a major public health concern [117]. These data are supported by a later study showing that pistachios are an effective alternative to a low-fat, high-carbohydrate food in order to improve postprandial glucose, insulin, and GLP-1 response in women with gestational diabetes mellitus and with gestational impaired glucose tolerance [118]. 

## 3. Conclusions

The available scientific evidence on the health benefits related to nut consumption has indicated an effect on cardiovascular and chronic disease prevention, anti-inflammatory and oxidative stress reduction, as well as functional food properties. Although studies investigating the effect of nuts in counteracting the impact of brain aging are more limited and often controversial, the presence of mono- and poly-unsaturated fatty acids, together with minerals, vitamins, and polyphenols could play an important role on cognitive performance, with an impact on aging and neurodegeneration. Lutein in pistachios is also a novel phytochemical affecting cognitive function. 

## Figures and Tables

**Table 1 ijms-22-05960-t001:** Glucose modulation effects related to nut consumption.

Nuts & Intervention	Study Design	Study Population	Outcome	Reference
Almonds (60 g)	Randomized trial	Healthy subjects	↑ Serum concentration of protein thiol ↓ Postprandial glycaemia, insulinemia, satiety	[12]
Almonds (43 g/day)	Randomized controlled trial	Participants with increased risk for T2DM	↑ Satiety, MUFA, α-tocopherol ↓ Serum glucose postprandial glucose	[10]
Pistachio (57 g/day)	Randomized, controlled, crossover trial	Pre-diabetic subjects	↑ GLP-1, ↓ FBG, insulin, HOMA-IR, fibrinogen, oxidized-LDL, platelet factor 4 ↓ IL-6 mrna, resistin gene and glucose uptake (in lymphocytes)	[8]
Pistachio (25 g twice/day)	Randomized, controlled, crossover trial	Patients with T2DM	↓ FBG and HbA1c	[9]
Walnuts (30 g/day)	Randomized, controlled, trial	Overweight adults with T2DM	↓ Fasting insulin levels, body weight (3–6 months)	[11]

↑: increase; ↓: decrease.

**Table 2 ijms-22-05960-t002:** Body weight management effects related to nut consumption.

Nuts & Intervention	Study Design	Study Population	Outcome	Reference
Nut-enriched diet	Meta-analysis	Participants with Mets and overweight/obesity	↓ Mets, overweight and or obesity, Body weight, BMI, and WC	[20]
Almonds (15% of diet energy)	Randomized, controlled, trial	Overweight and obese adults	↓ Total and truncal fat, diastolic BP	[21]
Almonds (28 g/day)	Randomized trial	Overweight and obese adults	↓TG, TC, TC/HDL-C (6 months)	[17]
Peanuts (89 ± 21 g/day)	Crossover intervention study	Healthy, normal-weight participants	↑ Satiety value No changes in energy intake and body weight	[22]

↑: increase; ↓: decrease.

**Table 3 ijms-22-05960-t003:** Cardiovascular disease prevention and serum lipid effects related to nut consumption.

Nuts & Intervention	Study Design	Study Population	Outcome	Reference
Nut intake	Meta-analysis of prospective studies	Adult populations	↓ Risk of cardiovascular disease, total cancer, all-cause mortality	[41]
Almonds/pistachios/walnuts/hazelnuts (fom 37 to 128 g/day)	Systematic review and meta-analysis	Adult population	↑ Endothelial function (walnuts)	[53]
Almonds (42.5 g)	Randomized, controlled, crossover trial	Adult individuals	↓ Non-HDL-C LDL-C TC/HDL-C, LDL-C/HDL-C, apob/apoa1, abdominal and leg fat mass, waist circumference	[15]
Whole roasted almonds as snacks	Randomized, controlled, paraller-arm trial	Adult individuals	↑ Endothelium-dependend vasodilation, ↓ LDL-C, no changes in liver fat and other risk factors	[54]
Pistachio (from 65 to 75 g/day)	Randomized trial	Healthy subjects	↑ HD, AOP and AOP/MDA ↓ TC, MDA, TC/HDL and LDL/HDL	[18]
Pistachio (57 g/day)	Randomized, crossover, controlled trial	Prediabetic individuals	↓ Sldl-P, non-HDL-P, HDL-P size	[16]
Cashews (28–64 g/day)	Randomized, controlled, crossover trial	Hypercholesterolemic adults	↓ TC, LDL-C, non-HDL-C, TC/HDL-C	[14]
Nuts intake (50–100 g/d)	Systematic review	Adult individuals (healthy/hypercholesterolemic/hyperlipidemic/ diabetic)	↓ TC, LDL-C	[19]
Pistachio diet (8.1%) Or mixed nut diet (7.5%)	Animal study	Rats	Antioxidant, antiinflammatory, and hypolipidemic effects	[13]

↑: increase; ↓: decrease.

**Table 4 ijms-22-05960-t004:** Effect on inflammation and oxidative stress related to nut consumption.

Nuts and Intervention	Study Design	Study Population	Outcome	Reference
Almonds (75% energy intake) Walnuts (75% energy intake)	Randomized crossover trial	Healthy subjects	↑ Polyphenol concentration ↑ Antioxidant capacity ↓ Lipid peroxidation	[40]
Nuts intake	Randomized trial	Overweight and obese stable coronary artery disease individuals	↓ ICAM-and IL-6	[29]
Almonds (56 g/day)	Randomized, controlled crossover trial	Subjects with T2DM and mild-hyperlipidemia	↑ LDL resistence vs Cu^2+^ ↓ IL-6, CRP, TNF-α, PC	[38]
Hazelnuts (30 or 60 g/day)	Randomized, controlled trial	Overweight and obese individuals	↓VCAM-1 (60-g/d)	[37]
Cashew nuts (100 mg/kg)	Animal study	Rats with osteoarthritis	↑ Antioxidant and anti-inflammatory effects	[32]
Cashew nuts (100 mg/kg)	Animal study	Rats with intestinal I/R injury	↑ Antioxidant and anti-inflammatory effects.	[33]
Cashew nuts (100 mg/kg)	Animal study	Rats with colitis.	↑ Antioxidant and anti-inflammatory effects	[35]
Cashew nuts (100 mg/kg)	Animal study	Rats with paw edema	↑ Antioxidant and anti-inflammatory effects	[30]
Cashew nuts (100 mg/kg)	Animal study	Rats with pancreatic and lung injury	↑ Antioxidant and anti-inflammatory effects	[31]
NP or RP30 (mg/kg)	Animal study	Rats with paw edema	↑ Antioxidant and anti-inflammatory effects	[36]
Nuts intake	Review	Study in vitro and in vivo	↑ antioxidant effects	[34]
Almond skin (30 mg/kg)	Animal study	Rats with spinal cord injury	↑ Anti-inflammatory effects	[39]

↑: increase; ↓: decrease.

**Table 5 ijms-22-05960-t005:** Functional food properties.

Nuts & Intervention	Study Design	Study Population	Outcome	Reference
Finely ground almonds or defatted finely ground almonds	In vitro gastric and duodenal digestion	Mixed fecal bacterial cultures	↑ bifidobacteria and *Eubacterium rectale* (FG )	[25]
Natural almond skin or blanched almond skin	In vitro gastric and duodenal digestion	Mixed fecal bacterial cultures	↑ bifidobacteria and *Clostridium coccoides/Eubacterium rectale* group	[27]
Almonds or pistachios (1·5 or 3 servings/d)	Randomized, controlled, crossover trial	Healthy subjects	↑ butyrate-producing bacteria (pistachio) ↓ lactic acid bacteria (pistachio)	[24]
Roasted almonds (56 g) Or almond skins (10 g)	Randomized trial	Healthy subjects	↑ *Bifidobacterium* spp. and *Lactobacillus* spp. ↓ *Clostridum perfringens*	[23]

↑: increase; ↓: decrease.

**Table 6 ijms-22-05960-t006:** Cognitive performance related to almond consumption.

	Study Population	Outcome	Reference
Randomized trial	Middle age to older people	↑ Alpha-tocopherol ↑ Memory and learning	[64]
Randomized trial	Older overweight adults	No changes in mood and cognitive performance	[65]
Randomized trial	Overweight and obese adults	↑ Post-lunch dip in memory No changes in cognitive performance	[66]
Randomized trial ^a^	Older adults with high vascular risk	↑ Cognition	[67]
Randomized trial ^a^	Older adults with high vascular risk	No differences	[68]
Randomized trial ^b^	Older healthy women	No differences	[69]
Cross-sectional ^b^	Older healthy adults	Lower risk of cognitive impairment	[70]
Prospective cohort ^c^	Older healthy adults	Lower risk of cognitive decline	[71]
Cross-sectional ^b^	Older healthy adults	No differences	[72]
Animal study	Healthy rats	↑ Memory retention	[73]
Animal study	Healthy rats	↑ Memory retention ↑ Scopolamine-induced amnesia	[74]

^a^: MED-DIET plus almonds and hazelnuts; ^b^: study administering total nuts; ^c^: study administering walnuts, almonds, hazelnuts and peanuts. ↑: increase.

## Data Availability

Not applicable.

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
