# Peer review of "Health Benefits Related to Tree Nut Consumption and Their Bioactive Compounds"

_ijms, 2021, doi:10.3390/ijms22115960_

Round 1

Reviewer 1 Report

1. The authors present some interesting information, but I feel confused about the logic and the focus of this manuscript. I was expecting to see specific compounds that are beneficial to human health. Instead, the authors talk a lot more about controlling NCDs than focusing on nuts. What is more, through the whole text the reader is not actually sure which nuts are the authors targeting.

2. Talking about different diets like the Mediterranean one, does not prove their point at all.

3. Revise sentence in lines 42-44

4. I suggest a reorganisation of the columns in Table 1. Since nuts are the subject of this manuscript, they should be placed in the first column.

This manuscript needs major improvements in order to meet the expectations from the title.

Author Response

The authors would like to thank the reviewers for their comments and very much appreciate the time to review the work. We have carefully reviewed the observations and have revised the paper accordingly. We hope these changes meet with your approval.

  1. More information has been added in order to highlight the specific compounds responsible for the health outcomes identified with nut consumption in each of the 5 main categories reported (glucose modulation, body weight management, cardiovascular disease prevention, inflammation and oxidative stress, functional food properties). However considering that the special issue to which the review has been submitted is “Physiological or Pathological Molecular Alterations in Brain Aging”, we focused on all the health properties of nuts  including  their positive effects on NCDs, which have been widely discussed in consideration of the fact that to date little is known about the effects of nuts on cognitive performance.
  2. The review does not focus on the different diets; in the manuscript the diet is just an element in a context. We did not consider necessary to explore this aspect further considering the main topic of the review is the all the health properties of nuts.
  3. we revised the sentence in line 42-44
  4. the table has been reorganized as suggested

Reviewer 2 Report

  • The review is well written and everything is comprehensible.
  • Table1 is of great significance, while authors should split it, in more tables regarding each of the corresponding effect and/or action, in order to become more understandable.

Author Response

The authors would like to thank the reviewers for their comments and very much appreciate the time to review the work. We have carefully reviewed the observations and have revised the paper accordingly. We hope these changes meet with your approval.

According to reviewer suggestion the table has been reorganized as suggested

Round 2

Reviewer 1 Report

Changes in the tables have been made, and the authors have a strong opinion on the manuscripts inital content. Still, I would suggest a modification in the title to make it more comprehensive.

Author Response

According to reviewer suggestion we have modify the title as: "Health Benefits Related to Tree Nut Consumption and Their Bioactive Compounds". We hope that the change is suitable, otherwise if the reviewer thinks that are necessery further changes, please suggests us more carefully the changes to perform.